# Head-Mounted Display for Clinical Evaluation of Neck Movement Validation with Meta Quest 2

**DOI:** 10.3390/s23063077

**Published:** 2023-03-13

**Authors:** Manuel Trinidad-Fernández, Benoît Bossavit, Javier Salgado-Fernández, Susana Abbate-Chica, Antonio J. Fernández-Leiva, Antonio I. Cuesta-Vargas

**Affiliations:** 1Grupo de Investigación Clinimetría, Departamento de Fisioterapia, Universidad de Málaga, 29071 Málaga, Spain; 2Instituto de Investigación Biomédica de Málaga y Plataforma en Nanomedicina (IBIMA), Plataforma Bionand, 29590 Málaga, Spain; 3ITIS Software, Departamento de Lenguajes y Ciencias de la Computación, Universidad de Málaga, Andalucía Tech, 29071 Málaga, Spain; 4Departamento de Expresión Gráfica, Diseño y Proyectos, Escuela de Ingenierías Industriales, Universidad de Málaga, 29071 Málaga, Spain; 5School of Clinical Sciences, Faculty of Health, Queensland University of Technology, Brisbane, QLD 4000, Australia

**Keywords:** virtual reality, Meta Quest 2, neck, rehabilitation, validation

## Abstract

Neck disorders have a significant impact on people because of their high incidence. The head-mounted display (HMD) systems, such as Meta Quest 2, grant access to immersive virtual reality (iRV) experiences. This study aims to validate the Meta Quest 2 HMD system as an alternative for screening neck movement in healthy people. The device provides data about the position and orientation of the head and, thus, the neck mobility around the three anatomical axes. The authors develop a VR application that solicits participants to perform six neck movements (rotation, flexion, and lateralization on both sides), which allows the collection of corresponding angles. An InertiaCube3 inertial measurement unit (IMU) is also attached to the HMD to compare the criterion to a standard. The mean absolute error (MAE), the percentage of error (%MAE), and the criterion validity and agreement are calculated. The study shows that the average absolute errors do not exceed 1° (average = 0.48 ± 0.09°). The rotational movement’s average %MAE is 1.61 ± 0.82%. The head orientations obtain a correlation between 0.70 and 0.96. The Bland–Altman study reveals good agreement between the HMD and IMU systems. Overall, the study shows that the angles provided by the Meta Quest 2 HMD system are valid to calculate the rotational angles of the neck in each of the three axes. The obtained results demonstrate an acceptable error percentage and a very minimal absolute error when measuring the degrees of neck rotation; therefore, the sensor can be used for screening neck disorders in healthy people.

## 1. Introduction

Neck pain plays an important role in people because of the high prevalence and the great disability it entails in daily life activities, as the neck has an important involvement in these activities such as head positioning and turning the head with any perceived stimulus [1]. Although classification of neck pain can be extensive and difficult to identify, these usually originate from two main reasons: (i) the pain is related to a purely mechanical cause as a result of a trauma or maintained or overuse postures; or (ii) the cause is not due to damaged tissue but to other factors such as psychosocial ones, known as non-specific neck pain [2,3]. According to the type of injury, the clinician or physiotherapist can determine a specific treatment. For instance, in case of mechanical reason, sessions should focus on resolving the tissue problem, whereas for non-specific pain, interventions will concentrate on patient education, physical exercise, and manual therapy [4,5]. The last recommendations from the International Collaboration on Neck Pain supported the assessment of the range of movement restrictions in people with neck-related musculoskeletal conditions [6]. There are several action guides to assess specific neck pathologies, such as cervical dystonia [7], which are based on clinical features observed by the rater. Observation by a rater can be subjective, and objective quantifiable data are necessary. Several types of reliable devices have been used to recognize neck movement, but constant calibration and burdensome use make them difficult to transfer to the clinic [8]. Therefore, a good assessment of the neck is primordial, and it depends on the medical history and physical examination through palpation, as well as on different tests where the use of measuring devices is essential [1]. 

Virtual reality (VR) is defined as a digital experience in which the subject interacts with an artificial environment [9]. Currently, the evolution of this technique and the development of new devices have created different types of VR—non-immersive, semi-immersive, and immersive—and the main difference between them is the degree of interaction with the artificial environment and the real world [10,11]. While non-immersive and semi-immersive offer computer-generated environments with a not-fully absence of physical surroundings, immersive VR (iVR) offers a greater disconnection from the real world, as well as further interaction with the fictional world through the use of VR head-mounted displays (HMD) [10,11]. Recent research in the health field aims to employ such a technology as a therapeutic tool thanks to its accessibility, adaptability, ease of use, and affordability [9,11,12]. iVR is an important tool to improve the acquisition of motor skills better than conventional screens [13] and decreases the severity of symptoms such as anxiety and depression, fatigue, and pain [14]. In addition, it helps to increase adherence to treatment because of the playful and motivational components, as well as the different feedback it provides to the patient [11], the different types of stimuli, making the patient focus on the task performance in the virtual environment [9]. 

Although the use of HMD in neck assessment seems promising, research is scarce with limited evidence [12]. Currently, the best-selling HMD sensor is the Meta Quest 2 (Meta Platforms) because of its quality, affordability compared to the competition, and its use without cables, which offers freedom of movement to its users [15]. The main advantages of this device are: it is a wireless standalone device so it has an independence of a main CPU and external tracking system; the application in the 6 degrees of freedom; it has a good screen and in-self tracking; and the large amount of accessible content and applications [16,17]. Thanks to the use of their cameras, the device provides an inside-out tracking system for hand or controller recognition, making easier the naturally interaction in the virtual environment [16,17]. Other current devices can have better resources and excellent reliability but the high price of these systems makes the Meta Quest 2 an affordable solution to dispose of them in consultation with patients. Nevertheless, the VR headset sensor has not yet been validated for clinical use.

Due to the high prevalence of neck pain, there is a need for research into new, more accurate, and faster assessment tools [3,12]. Therefore, the aim of this study is to verify the validity of the Meta Quest 2 HMD system as a method for neck-movement assessing in healthy subjects, before extending its use to people with neck disorders. The study expectations were that iVR Meta Quest 2 HMD was a valid and reliable method for neck assessment, obtaining kinematic parameters that match real values to potential usage as an assessment method.

## 2. Related Work

Virtual reality and other technologies have the potential to assess cervical range of motion (ROM) considering three axes of movement such as flexion-extension, rotation, and lateral flexion, as well as speed, balance, and reaction time [12,18]. There are many studies and commercial applications that already use iRV to assess and treat neck pathologies such as neck pain and vestibular problems [19]. The features of iRV favor several symptoms and signs of neck pain disorders such as kinesiophobia [9] and impairments [20]. Regarding the assessment, Kieper et al. 2020 proved Oculus Rift HMD (Meta Platforms) provided more accurate information about neck range of motion than non-immersive techniques [21]. In the same vein, the same headset was used to evaluate neck movement in elderly people during driving [22]. The authors showed how young people had more functional motion than elderly people during an immersive driving experience [22].

Several studies validating HMD systems have been recently published in the scientific literature. The Meta Quest 2 controller, not the headset, was previously validated showing a mean absolute error of 13.52 mm and a maximum mean absolute error in rotations of 1.11° [23]. Regarding the validation of an HMD headset in neck movement, Oculus Rift presented a mean error between 3.9° and 9.5° in all the axes of cervical spine mobility [24]. Sarig-Bahat et al. 2009 analyzed the I-Glasses HRV Pro (Virtual Realities) HMD system with the same purpose [25]. They obtained mean differences between 7.2° and 16.1° in flexion-extension and axial rotation, respectively. Another study showed excellent intra- and inter-rater results (ICC = 0.83–0.97) in the cervical movement assessment with the HTC Vive Pro Eye [26]. 

Other devices have been used to analyze neck movement in the literature. The most-used device was the inertial measurement units (IMU), proving good and similar results. The validated IMUs during these years were AHS EBIMU-9DoF (E2BOX) [27], IMU Lynx (DyCare) [28], and Shimmer 3 (Shimmer Sensing) [29]. The IMUs inside of smartphones provide sufficient validity results but low quality of the evidence [30]. Video camera systems have been also checked for the same goal. The Veloflex system is a optoelectronic one-camera system with markers that showed excellent results in the validation with a mean difference less than 1° [31]. On the other hand, the Microsoft Kinect camera and iPad Mini 4 (Apple) were used to assess neck range of motion, demonstrating the limitation that one camera with no references has to collect neck angles [27,32].

## 3. Materials and Methods

This section is devoted to explaining the methodology used to carry out this study. In Section 3.1 and Section 3.2, the experimental setup and procedure is described, respectively, and later, the analysis performed is in Section 3.3.

### 3.1. Experimental Setup

The Meta Quest 2 Virtual Reality HMD device (Meta Platforms, Inc., Menlo Park, CA, USA) was used in this study. The weight and dimensions of the HMD are 503 g and 22.4 × 45.0 cm, respectively. In addition, it is self-contained with a rechargeable 3640 mAh lithium-ion battery. The main important technical features of this device are an LCD display of 1832 × 1920 per eye with a display rate up to 90 Hz, a Qualcomm Snapdragon XR2 as CPU, 6 GB of RAM, 128 GB of memory, four infrared cameras, and built-in speakers [16]. Although two Meta manual controllers are incorporated into the system, their use was limited to access the application. The HMD contains a TDK ICM-42688-P (InvenSense Inc., San Jose, CA, USA) inertial measurement unit (IMU) inside of the headset that tracks 6 degrees of freedom to collect head movements.

The custom VR application developed by the authors of this paper was built with Unity game engine 2021.3.6f1 and Oculus XR Plugin software development kit (SDK) 3.0.2. The Oculus SDK provides the position and orientation of the head, and the data were saved into a JSON file on the headset hard-drive. Motion information for the HMD during the application was recorded with fixed intervals at 30 frames per seconds.

The criterion standard used in this study was an InertiaCube3 (Thales Intersense Inc., Billerica, MA, USA) IMU with dimensions of 26.2 × 39.2 × 14.8 mm and a weight of 17 g. It contained an inertial sensor (accelerometer, gyro, and magnetometer) with a 3-degree-of-freedom orientation tracking system—yaw, pitch, and roll—and with a sampling frequency of 180 Hz and accuracies of 1°, 0.25°, and 0.25°, respectively. The IMU was calibrated to the initial coordinate (0°, 0°, 0°) when the devices were on the participant and before the first measurement. ISense Software (Thales Intersense Inc., Billerica, MA, USA) was used to collect the data from the IMU, creating a txt file.

Figure 1 shows the placement of the IMU over the HMD and the coordinates reference system. A custom IMU holder was made by EVA foam paper, and it was created to reduce the possible noise caused by the movement of the sensor or the wire and helped in the placement because the upper side of the HMD device is a bit convex. The holder and the sensor were taped to the surface using several strips of tape over them to minimize more of the noise when testing.

### 3.2. Experimental Procedure

All the measurements were performed by two healthy volunteers that used the HMD with the IMU. The experiment was carried out in a large room in the Human Movement Laboratory of the Faculty of Health Sciences of the University of Malaga. The study obtained the approval from the Ethical Committee for Experimentation of the University of Malaga (64-2022-H) and the Commission of Investigation of the Faculty of Health Sciences (AntCue89).

The subject had to launch the application from the application menu to start the measurement protocol. The digital experience took place in a simulated gym facility with a front mirror to observe the virtual avatar mimicking the user’s movements as a visual guide. The starting position (0°, 0°, 0°) for both devices was set before every measurement and the volunteer was facing forward without any type of rotation, flexion, or lateralization of the head. The subject began with a calibration phase by performing a maximum neck movement looking to the left and right side (neck rotation), looking up and down (neck flexion), and leaning left and right (neck lateralization). This calibration was carried out to inform the system of the subject’s maximum neck movement and was not collected as a study variable.

After this, the real measurements began by asking the subject to perform six neck movements (rotation, flexion, and lateralization on both sides) at an intermediate orientation and at the maximum ROM recorded during calibration phase, with a total of 12 different head orientations. The decision of adding an intermediate orientation in the ROM was to measure the ability of detecting angular movement in addition to the maximum ranges of motion. To make sure that the participants oriented their heads at the desired angles for a controlled measurement, the activity proposed a system of targets that the participants must hit. To help the subject’s visual biofeedback, a system of cubes was designed to help mark positions in all axes, following a preliminary study about the recommendation of using geometric figures [33]. By rotating the head, the user manipulates a cube, with an arrow to show the vertical, that must be placed inside the target, which is a bigger version of this cube. To complete the task, the subject must maintain the cube within the target for 2 s; this ensures a controlled movement. Once the task is completed, the user must look straight again and maintain the stance for 2 more seconds before the target appears in the next position. In addition to this mechanism, the application provided an animation of a virtual puppet and messages such as “Look to the right”, “Look up”, or “Look forward again” to indicate the next move. Figure 2 presents an example of how the application works.

Figure 3 shows the different movements with the specific axis and direction. The 12 different head orientations in the following order were: Yaw axis: left neck rotation (look left) of 45° and maximum ROM, and right neck rotation (look right) of 45° and maximum ROM;Pitch axis: upward neck flexion (look up) of 20° and maximum ROM, and downward neck flexion (pitch axis) of 20° and maximum ROM;Roll axis: left neck lateralization (lean left) of 15° and maximum ROM, and right neck lateralization (lean right) of 15° and maximum ROM.

There was no time limit to complete the tasks. The volunteers repeated 3 times the measurement protocol with 3 min of rest in between, for a total of *n* = 143 (12 measurements × 12 head orientations × 3 repetitions).

### 3.3. Data Analysis

HMD and IMU angular movement data were processed and compared. Resampling at the minimum sampling rate was performed. Since the data from the IMU were obtained via the proprietary software ISense with their own timestamp reference, the authors developed an internal tool with the Unity game engine to read data from both VR and IMU saved files. The internal tool moves the curve from IMU data along the timeline to visually overlap both curves. In addition, it corrects the possible small offset of the devices caused by the minimum errors they may have. Once the data were synchronized, the angles were saved in a comma-separated values (csv) file for each iteration and position.

Descriptive information of the 12 positions was presented as mean and standard deviation (SD). Data were averaged during the 2 s of recording in the exact neck movement position, so the mean measured position of each angular displacement was performed. All analyses were done using SPSS version 25 software (IBM, Armonk, NY, USA). Several parameters were calculated to analyze the accuracy of the HMD: absolute error, criterion validity, and agreement. These results were presented individually per movement and axis, besides to average the results of all the variables.

#### 3.3.1. Absolute Error

The mean absolute error (MAE) is the absolute value of the difference between an observed value of a quantity and the true value [34]. Then, the absolute difference between the HMD and IMU values was analyzed. The MAE of the angular movements were calculated with Equation (1):(1)$$\mathrm{MAE}=\left|\mathrm{Mean}\;\mathrm{HMD}\;\mathrm{movement}\;-\;\mathrm{Mean}\;\mathrm{IMU}\;\mathrm{movement}\right|$$

The MAE percentage (%MAE) was also calculated to clearly evaluate the absolute error with Equation (2):(2)$$\%\mathrm{MAE}=\left(\frac{\mathrm{MAE}}{\mathrm{Mean}\;\mathrm{IMU}\;\mathrm{movement}}\right)\cdot 100$$

#### 3.3.2. Criterion Validity and Agreement of the HMD-IMU

The criterion validity is the degree to which the scores of a device are an adequate reflection of a gold standard [35]. The criterion validity was calculated by comparing the angular displacement of the HMD and the IMU by bivariate correlation with the Pearson (r) correlation coefficient. The criteria to evaluate the correlation were r < 0.49 (poor), r = 0.50–0.74 (moderate), and r > 0.75 (strong) [36]. Additionally, Bland–Altman plots were created for the 12 positions to show the degree of agreement of the measure tools by defining the 95% limits of agreement (LoA) [37]. The LoA Equation (3) between the HMD and IMU is: LoA = Mean Difference ± 1.96·SD(3)

## 4. Results

Descriptive results of 12 positions from the HMD and the IMU are presented in Table 1. Table 2 reports the absolute and percentage errors and the correlations for all rotational movements made around the three axes. The mean absolute errors were similar between positions and not higher than 1° (average = 0.48 ± 0.09). The maximum mean absolute error was found in the yaw rotation to the left side at maximum ROM (0.67 ± 0.46°). The percentage error was not higher than 3.11%, specifically in the roll lean movement to the left at 15°. The average percentage of the rotation movement was 1.61 ± 0.82%. Regarding the criterion validity, the midpoint movements obtained a correlation between 0.70 and 0.96. The maximum values showed an excellent correlation (r = 0.98–0.99). The HMD and IMU systems have good agreement according to the Bland–Altman analysis (Figure 4). The highest mean difference was −0.59° in yaw rotation to the left at maximum ROM. The full database of this study is provided in Appendix A
Appendix A. 

## 5. Discussion

This study analyzes the absolute error, criterion validity, and agreement of the Meta Quest 2 during neck movement. Overall, the findings showed that Meta Quest 2 provides a validated measurements of the ROM of all movements in the three axes since the absolute error between the HMD and the gold standard was very small (less than 1°). The excellent correlations and the small LoAs validate the use of this HMD system to measure and assess neck mobility. This finding means that Meta Quest 2 can be considered as an alternative to other analogue and less-objective assessment methods with worse validity [38].

Meta Quest 2 has shown a very constant error in all movements (0.32–0.67°), which influenced the %MAE. Due to this constant error, we found that movements with fewer degrees such as lean left and right (roll axis) at 15° and look down and up (yaw axis) at 20° obtained the highest percentage in this analysis (2.04–3.11%). Therefore, this percentage was acceptable because the increase is directly correlated with grades and not with another external factor. Regarding other HMD systems that analyze neck movement, the information is limited because of the novelty of this tool and the fact that previous systems were more expensive and more oriented toward research and not to daily clinical practice. The Oculus Rift system, a previous version of Meta Quest 2 from the same company, showed acceptable absolute error results in the three maximum angular movements of the neck (2.3–5.4°) but worse than our study [24]. Being the same company, it may be that the Meta Quest 2 has evolved part of the previous system with a different IMU [39], and an improvement of certain plugins has improved the accuracy. Another different but important point is that this model of glasses could use external beacons, and they can influence the tracking of the device in space. The Meta Quest 2 is capable of tracking itself with the built-in sensors and inside-out tracking; therefore, it seems that the improvement in tracking compared to the previous generation is evident. In addition, the authors compared the Oculus Rift with an optoelectronic camera system where the reference system is totally different. In this validation of Meta Quest 2, we have tried to compare this system that is based on an embedded IMU with another IMU to analyze the same reference system. On the other hand, the mean difference of the I-Glasses HRV Pro was 7.2° for the flexion-extension (pitch axis) and 16.1° for the axial rotation (yaw axis) compared with an electromagnetic sensor [25]. This immersive virtual reality system was one of the first to market, so probably, the primitive device could have increased this significant error. The reference in that study was an electromagnetic sensor that had an excellent correlation and a small mean difference in analyzing cervical movement [40]. For that reason, this work does not represent the actual state of HMD devices, but it is relevant to take into consideration how previous authors tried to use this system to assess neck movement showing a low–moderate error. Much progress has been made in obtaining valid results of the headset movement in recent years, and this study with the Meta Quest 2 may be the beginning of future validation analyses of future versions of HMD systems.

In addition to these tools, it is relevant to compare the Meta Quest 2 with other IMU devices because of the previously mentioned component inside. The IMU Lynx (DyCare) and the IMU Shimmer3 (Shimmer Sensing) had similar results in their respective studies to measure neck motion. The correlations of the studies with the gold standard ranged from good to excellent (r = 0.65–0.95; r = 0.60–0.99) [29]. On the other hand, Ghorbani et al. 2020 also obtained similar results when validating the use of the IMU within an iPhone (r = 0.63–0.94) and a Samsung smartphone (0.45–0.91) [41]. The only study found that uses an IMU and has better results than the Meta Quest 2 was on the Wireless3 AHS EBIMU-9DoF (E2BOX), where the correlations were excellent in the three axes (r = 0.96–0.97), but the mean difference was worse than the HMD system (−7.0, −7.4, −7.5) [27]. These results suggest that the validation of IMUs to measure neck movement usually has similar data in general. The device with the best validation data for this objective was the Veloflex system, which obtained an excellent correlation (r = 0.96–0.99) and a mean difference similar to that of Meta Quest 2 (0.17–0.96°) [31]. This system can be a rival in terms of the precision of Meta Quest 2, but they are two totally different tools and the HMD system offers such a large resource capacity that it may be preferable to assess neck movement than the system with cameras, always taking into account the clinical context. We cannot ignore the influence that the weight and the placement of the HMD can have, which can modify the posture of the neck. These less heavy devices can give more biomechanical results since the distribution of forces is not altered with the VR system [42]. Despite this, early studies suggest that good neck muscles that can withstand this torque can make this device functional with neck movement [43]. In short, the use of the Meta Quest 2 virtual reality headset was justified according to the previous scientific literature based on other IMUs and camera systems.

### Strengths and Limitations

The validation of this tool allows that its use in rehabilitation is justified to find the real kinematics of the neck. The device could assess the movement impairments that have been proved using the kinematics such as greater neck flexion or reduced velocity [44]. This work is added to the validation of the controllers previously carried out by other authors [23]. Several studies and commercial applications that already used iRV to assess and treat neck pathologies in the Meta Quest 2 were not supported by validity studies as a tool. This work reduces a gap that exists and gives more relevance to the use because the clinical validity is more scientifically tested [19]. Another important contribution of these results is the validation of the Meta Quest 2 headset device. This HMD system is one of the best sellers currently for its affordable price and the good features it has. In addition, not only are the VR scenarios important, but this study opens the door to new approaches to the device with mixed reality through the passthrough camera that are not yet fully developed, causing several limitations [45]. It is a reference for the initiation of iVR by which clinicians can use these experiences with patients outside the laboratory because it is easy to use and does not require cables or external sensors like other systems [15]. Therefore, the validation of this device helps to create new experiences and studies that will be more transferable to patients with a more scientific basis. 

Despite these strengths, the study has several limitations. Not using patients with any neck pathology made the validation not so specific for neck rehabilitation. Despite this, the healthy young adult population has served to carry out a complete validation analysis on a device widely used not only in neck rehabilitation. In addition, the 12 positions analyzed may be scarce. The neck is a joint that does not have as much freedom in 9-DoF as the shoulder, for example. These positions were chosen to calculate the maximum neck ROM, which is essential for neck assessment, and an intermediate position with the aim of representing treatment programs with neck movement that do not reach maximum ROM. Finally, we cannot forget the possible added error that can be found in the use of the IMU and in the HMD-IMU synchronization. The reliability error of the same sensor, which is documented, and the application of signal synchronization could have influenced the result, and it is something that has been tried to be reduced with a standardization of the sensor and the measurement. Despite this, the results showed a total error that is small, so this external error is acceptable.

## 6. Conclusions

The HMD Meta Quest 2 was proved to be a valid system to measure the degrees of rotational movements of the neck in the three axes. The results provided show a very small absolute error and an acceptable error percentage to measure the degrees of neck rotation. Furthermore, the criterion validity between the HMD system and a comparator IMU was considered good. The precision of the measurements of neck movements made this system useful during rehabilitation assessment. The Meta Quest 2 virtual reality headset is presented as a feasible alternative to conventional motion analysis systems and rehabilitation tools because of its good adequation. Future studies should test new experiences where neck movement is the main component and analyze therapeutic interventions in neck pain disorders.

## Figures and Tables

**Figure 1 sensors-23-03077-f001:**
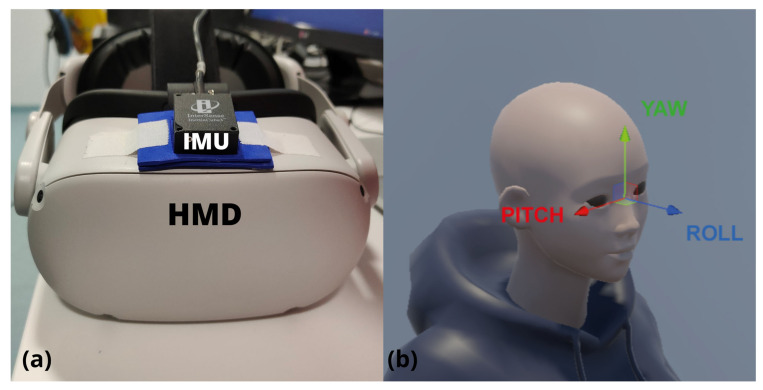
(**a**) Inertial measurement unit (IMU) placed over the head-mounted display (HMD); (**b**) coordinate reference system of the device in a test avatar.

**Figure 2 sensors-23-03077-f002:**
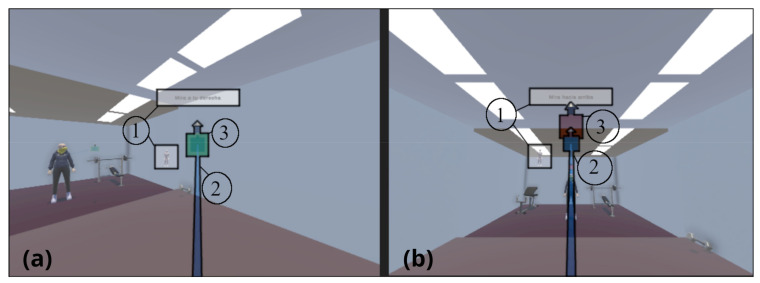
Examples of looking right (**a**) and looking up (**b**). Elements highlighted with number 1 are the instructions consisting of puppet animation and text output. Elements highlighted with number 2 are the visual feedback about user’s gaze. Finally, elements highlighted with number 3 are the targets to reach.

**Figure 3 sensors-23-03077-f003:**
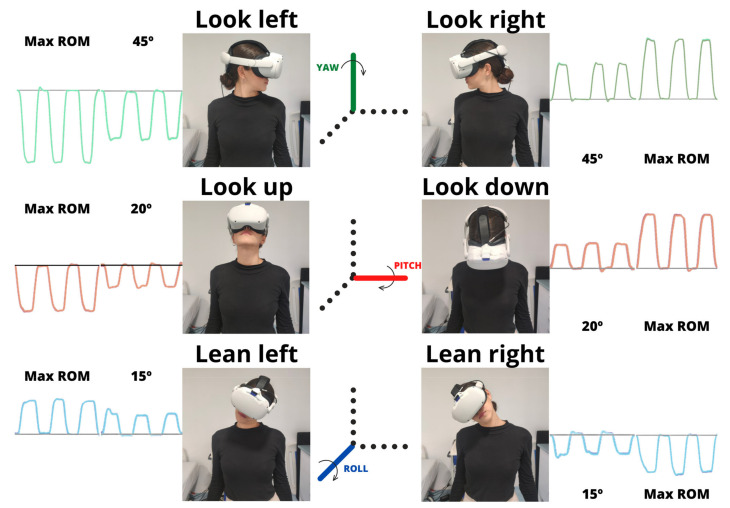
Subject performing the six maximum movements and the corresponding axis. Max ROM—maximum range of motion. The black lines represent 0° of the selected movement.

**Figure 4 sensors-23-03077-f004:**
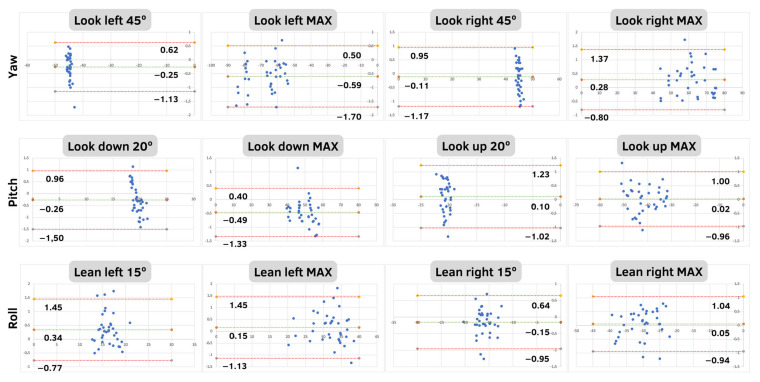
Bland–Altman plots to assess the agreement between HMD system and the IMU in all the positions and axes. The plots include the mean difference (green line) and limits of agreement (red lines).

**Table 1 sensors-23-03077-t001:** Descriptive outcomes of neck movement obtained from both measurement devices.

		HMD		IMU	
Axis		Mean (°)	SD	Mean (°)	SD
Yaw	Look left 45°	−44.98	0.53	−44.73	0.73
	Look left MAX	−67.58	9.93	−66.98	9.80
	Look right 45°	44.78	0.61	44.89	0.86
	Look right MAX	61.86	9.79	61.57	9.83
Pitch	Look down 20°	19.27	0.67	19.54	1.09
	Look down MAX	49.37	5.26	49.84	5.41
	Look up 20°	−20.56	0.68	20.66	0.78
	Look up MAX	−41.43	5.91	−41.45	5.95
Roll	Lean left 15°	16.29	1.89	15.95	1.94
	Lean left MAX	31.87	4.45	31.71	4.52
	Lean right 15°	−16.08	1.65	−15.92	1.64
	Lean right MAX	−30.14	4.71	−30.20	4.63

HMD, head-mounted system; IMU, inertial measurement unit, SD, standard deviation.

**Table 2 sensors-23-03077-t002:** Error and validity results in rotational movements comparing the HMD system and the IMU.

		Error		Validity
Axis		MAE (°)	%MAE (%)	r
Yaw	Look left 45°	0.38 ± 0.34	0.86 ± 0.77	0.78
	Look left MAX	0.67 ± 0.46	1.01 ± 0.71	0.99
	Look right 45°	0.45 ± 0.30	1.01 ± 0.69	0.78
	Look right MAX	0.49 ± 0.37	0.83 ± 0.62	0.99
Pitch	Look down 20°	0.58 ± 0.34	2.96 ± 1.77	0.85
	Look down MAX	0.54 ± 0.34	1.07 ± 0.69	0.99
	Look up 20°	0.49 ± 0.30	2.39 ± 1.48	0.70
	Look up MAX	0.39 ± 0.30	0.93 ± 0.74	0.99
Roll	Lean left 15°	0.48 ± 0.44	3.11 ± 2.85	0.95
	Lean left MAX	0.53 ± 0.40	1.72 ± 1.31	0.98
	Lean right 15°	0.32 ± 0.29	2.04 ± 1.84	0.96
	Lean right MAX	0.41 ± 0.29	1.42 ± 0.98	0.99
**Average**		0.48 ± 0.09	1.61 ± 0.82	0.91

MAE, mean absolute error; %MAE, percentage of mean absolute error; MAX, maximum range of motion; r, Pearson’s correlation.

## Data Availability

The datasets used and analyzed during the current study are available as Appendix A in this journal.

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
