# Peer review of "Head-Mounted Display for Clinical Evaluation of Neck Movement Validation with Meta Quest 2"

_sensors, 2023, doi:10.3390/s23063077_

Round 1

Reviewer 1 Report

The background provided for the study is not sufficient and relevant to the problem statement. The authors need to provide deeper and profound background study/review of literature.

Authors should provide details of HMD selection, its comparison with others available and justification of its utilization.

The contents be backed up with more up to date references and should come up with more technical discussion, contents, and implementation related details.

The actual data (atleast samples) should be provided and their values be reflected in the formulae provided.

The term population is not appropriate in the following statement in line No. 7 and 43.

1.       Neck disorders have a significant impact on the population due to its high incidence and the significant impairment it causes in day-to-day activities.

There are few other lines that needs rewording including the following:

1.       Physiotherapy treatment will depend on the type of injury …  line No. 51.

2.       There is an excessive use of will that needs to be avoided.

3.       Virtual reality (VR) could be defined as a digital experience in which the subject 59 interacts with an artificial environment [7] …  line No. 59.

4.       The angular movement MAE was 196 calculated following the formula .. line No. 197

Author Response

Point-by-point response to reviewer 1

Manuscript Number:   sensors-2227091

"Head-Mounted Display for clinical evaluation of neck movement validation with Meta Quest 2"

Dear Editor and reviewer,

Thank you very much for having in mind and review our manuscript. We are grateful for all the advice, comments and corrections suggested, which we believe are right to improve the quality of this paper.

Next, we will try to answer point by point to all your questions and suggestions, as well as specify the changes we have carried out. We highlighted in yellow all the changes in the manuscript.

  1. The background provided for the study is not sufficient and relevant to the problem statement. The authors need to provide deeper and profound background study/review of literature.

Authors: Thank you for your remark. We introduced more relevant references and points into the Background section. We expect the background is better under your criteria now.

  1. Authors should provide details of HMD selection, its comparison with others available and justification of its utilization.

Authors: Thank you for your comment. We included more solid details about the use of Meta Quest 2 in this study in the Background section. We expected that readers can notice why we selected this device among others. After this, readers also can compare the validity results specifically in the Discussion section.

  1. The contents be backed up with more up to date references and should come up with more technical discussion, contents, and implementation related details.

Authors: Thank you for your suggestion. We included more updated references and made more technical and implementation insights in the Discussion section.

  1. The actual data (at least samples) should be provided and their values be reflected in the formulae provided.

Authors: Thank you for your suggestion. We did not fully understand what you meant with the suggestion. Did you suggest uploading the database to a file repository? If it does, we agree with that and we will upload the database as a supplementary material.

The term population is not appropriate in the following statement in line No. 7 and 43.

  1. Neck disorders have a significant impact on the population due to its high incidence and the significant impairment it causes in day-to-day activities.

There are few other lines that needs rewording including the following:

  1. Physiotherapy treatment will depend on the type of injury … line No. 51.
  2. There is an excessive use of will that needs to be avoided.
  3. Virtual reality (VR) could be defined as a digital experience in which the subject 59 interacts with an artificial environment [7] … line No. 59.
  4. The angular movement MAE was 196 calculated following the formula .. line No. 197

Authors: Thank you for your recommendations. We rewrote all the sentences that you suggested.

  • Neck pain plays an important role in people.
  • According to the type of injury, the physiotherapist will determinate an specific treatment.
  • For instance, in case of mechanical reason, sessions should focus on resolving the tissue problem.
  • Virtual reality (VR) is defined as a digital experience where the subject interacts with an artificial environment.
  • The MAE of the angular movements were calculated with the formula (1).

Reviewer 2 Report

ABSTRACT

Too lengthy; consider reducing it's size (many of the details can be found in the Introduction and Methodology sections).

KEYWORDS

consider removing 'immersive' as it is a redundant term.

INTRODUCTION

There are 2 types of VR: semi-immersive and immersive; non-immersive VR does not exist. What to do mean by non-immersive VR? Are you referring to Augmented Reality or Mixed Reality? 

It is not clear why did the authors consider (immersive) VR since the purpose of this work was to validate the built-in IMUs of the headset. The same study could be performed in Video See-Through AR (Passthrough). Please discuss why.

Do what extent does the weight of the HMD device influence the movement performance? Please add a short discussion. 

In the Results section, please add a table to compare the values between the Oculus Rift HMD [11], HTC Vive Pro Eye [12] and the Meta Quest 2.

2.1 EXPERIMENTAL SETUP

Please also add the weight of the device (in grams).

The sampling frequency of the Quest (30 Hz) and the InertiaCube3 (180 Hz) are very different. Please comment.

The Figure 1 showed the --> Figure 1 shows the

Add a few more details about the custom IMU holder (what is it made of, how is the IMU fixed to it, etc).

Figure 1 could be improved as the roll axis is completely misaligned with the Anterior-Posterior axis of the headset. Try to represent a reference system that is really aligned with the headsets reference system. Consider doing so by rendering a line drawing instead of a photo.

2.2 EXPERIMENTAL PROCEDURE

There is no need for the University to be anonymized ("University of XXX"). 

Please specify how was the virtual mirror implemented? Virtual mirrors add extra computational burden to the application (computation of light rays). Did this not affect the application's run-time? Were there no rendering delays?

Figure 2 is hard to understand (its too dark, the color contrast is poor, the cube is not that visisble). You need to improve the figure. Also add labels and arrows to signal the important assests in each scene as this helps reading the figure content. Moreover, all of the decisions regarding visual content are not justified. Why a gym? Why does the avatar have an akward mask to cover his/hers mouth/nose? Is the user constantly seeing the head-gaze ray? I also feel that literature on visual biofeedback is missing. Missing reference: D.S. Lopes, A. Faria, A. Barriga, S. Caneira, F. Baptista, C. Matos, A.F. Neves, L. Prates, A.M Pereira, H. Nicolau, Visual Biofeedback for Upper Limb Compensatory Movements: A Preliminary Study Next to Rehabilitation Professionals, In Eurographics 2019 - Posters Track, DOI: 10.2312/eurp.20191139

Only 2 healthy participants that repeated 3 times each angular motion is not enough (6 samples per head orientation). Such a sample requires at least 15 samples (either more participants and/or more repititions) per each of the 12 head orientations. In fact, I did not understand the math because the authors then mention "The volunteers repeated 6 times ...": the sampling count is confusing and needs to be crystal clear!

Figure 3: please explain what is the meaning of the horizontal black lines; does the user also see the graphs? please clarify; and again, the color contrast is poor (colored graphs against a bray-blue gradient); This figure needs to be revisited and improved.

Figure 3: Max ROM, Maximum Range of Motion. --> Max ROM - Maximum Range of Motion.

2.12 Data analysis --> sub-section 12 ?!?; please update the section numbers throughout the manuscript.

All equations need to be numbered and all equations need to be formated ... they seem that they were writen inline and not with an Equation editor (e.g., the division line should cross the entire right side of the formulas and be represented by a modest '/')

limits of agreement (LoA) --> Limits of Agreement (LoA)

Author Response

Point-by-point response to reviewer 2

Manuscript Number:   sensors-2227091

"Head-Mounted Display for clinical evaluation of neck movement validation with Meta Quest 2"

Dear Editor and reviewer,

Thank you very much for having in mind and review our manuscript. We are grateful for all the advice, comments and corrections suggested, which we believe are right to improve the quality of this paper.

Next, we will try to answer point by point to all your questions and suggestions, as well as specify the changes we have carried out. We highlighted in yellow all the changes in the manuscript.

  1. ABSTRACT: Too lengthy; consider reducing it's size (many of the details can be found in the Introduction and Methodology sections).

Authors: Thank you for your recommendation. We have reduced the abstract changing information from the Introduction and Methodology.

  1. KEYWORDS: consider removing 'immersive' as it is a redundant term.

Authors: Thank you for your suggestion. We removed “immersive” from the Keywords terms.

  1. INTRODUCTION: There are 2 types of VR: semi-immersive and immersive; non-immersive VR does not exist. What to do mean by non-immersive VR? Are you referring to Augmented Reality or Mixed Reality?

Authors: Thank you for your comment. We know that there is a huge taxonomy mess in this field. The classification about Augmented Reality, Mixed Reality and Virtual Reality was described by Milgram and Kishino (1994) where they presented a continuum from the reality to virtual environment experiences. The types of virtual reality were described by Mujber et al. (2004) and followed by Muhanna (2014).  We included a brief context of this taxonomy and a definition of non-immersive and semi-immersive VR.

  1. It is not clear why did the authors consider (immersive) VR since the purpose of this work was to validate the built-in IMUs of the headset. The same study could be performed in Video See-Through AR (Passthrough). Please discuss why.

Authors: Thank you for your recommendation. Although for this specific study immersive environment is not necessary, the next idea is to implement an immersive assessment tool. For this reason, we decided to validate the technology reproducing a similar digital experience. Additionally, the passthrough camera of the Quest II has not a good quality and the experiences loses immersion. We agree that the option of use a Video See-Through AR could be very interesting in following updates, in the high-level HMD or maybe in the next generation (Quest III). We included a brief discussion about it in the Discussion section.

  1. Do what extent does the weight of the HMD device influence the movement performance? Please add a short discussion.

Authors: Thank you for your remark. We agree that the device can influence the movement and the neck posture because the HMD becomes a new part of the head, so the weight and the placement create an alteration in the load distribution. As Ito et al. 2021 suggested, the loss of mass or strength in the neck muscles can be improved with the HMD weight or, in our opinion, a traditional rehabilitation exercises before the HMD intervention if the patient has pain or discomfort symptoms. We include a brief discussion about it in the Discussion section.

  1. In the Results section, please add a table to compare the values between the Oculus Rift HMD [11], HTC Vive Pro Eye [12] and the Meta Quest 2.

Authors: Thank you for your recommendation. In our opinion, it is not possible to create a table to compare those results between devices because only the Oculus Rift and Meta Quest 2 were validated following the same analysis (absolute error). In the Discussion section, we provided all the comparations in the literature about HMD and other assessment devices. If you recommend us to create a table with all these results for the Discussion section, we will create one.

  1. 2.1 EXPERIMENTAL SETUP: Please also add the weight of the device (in grams).

Authors: Thank you for your remark. We included that the device weight 503 g.

  1. The sampling frequency of the Quest (30 Hz) and the InertiaCube3 (180 Hz) are very different. Please comment.

Authors: Thank you for your comment. The refresh rate of the Quest 2 can reach up to 120 Hz. In the case of application of our study, a data collection rate of 30 frames is set to ensure the stability at which the samples are taken, without depending on the device performance. In this particular, the InertiaCube3 samples are interpolated (6 more samples per second) and the HMD matches each sample taken by the Quest with the closest IntertiaCube sample.

  1. The Figure 1 showed the --> Figure 1 shows the

Authors: Thank you for your recommendation. We improved the sentence following your comment.

  1. Add a few more details about the custom IMU holder (what is it made of, how is the IMU fixed to it, etc).

Authors: Thank you for your suggestion. We prepared a holder made by EVA foam paper to place the sensor. Thanks to the holder, the sensor minimized the small movements of the sensor or the wire during the tests and helped to the placement because the upper side of the device (“the googles”) is a bit convex. The holder was taped to the device and the sensor placed inside the holder was further taped to minimize possible micro-movements when testing. We included all this new information in the Methods part.

  1. Figure 1 could be improved as the roll axis is completely misaligned with the Anterior-Posterior axis of the headset. Try to represent a reference system that is really aligned with the headsets reference system. Consider doing so by rendering a line drawing instead of a photo.

Authors: Thank you for your suggestion. We created a better figure with an aligned reference. We included a screenshot of the app avatar with the axes of rotation aligned to the reference system in Unity.

  1. 2.2 EXPERIMENTAL PROCEDURE: There is no need for the University to be anonymized ("University of XXX").

Authors: Thank you for your comment. We wrote the correct name of the University.

  1. Please specify how was the virtual mirror implemented? Virtual mirrors add extra computational burden to the application (computation of light rays). Did this not affect the application's run-time? Were there no rendering delays?

Authors: Thank you for your recommendation. The virtual mirror was simulated with a second camera and the render to texture technique in order to reduce computational costs.

  1. Figure 2 is hard to understand (its too dark, the color contrast is poor, the cube is not that visisble). You need to improve the figure. Also add labels and arrows to signal the important assests in each scene as this helps reading the figure content. Moreover, all of the decisions regarding visual content are not justified. Why a gym? Why does the avatar have an akward mask to cover his/hers mouth/nose? Is the user constantly seeing the head-gaze ray? I also feel that literature on visual biofeedback is missing. Missing reference: D.S. Lopes, A. Faria, A. Barriga, S. Caneira, F. Baptista, C. Matos, A.F. Neves, L. Prates, A.M Pereira, H. Nicolau, Visual Biofeedback for Upper Limb Compensatory Movements: A Preliminary Study Next to Rehabilitation Professionals, In Eurographics 2019 - Posters Track, DOI: 10.2312/eurp.20191139

Authors: Thank you for your remarks. We improved the figure with an example and some elements highlighted. Regarding the gym and the avatar, we decided to use a neutral environment such as gym club so that the participants feel comfortable to perform few exercises. Since participants could be from different genders, we decided to use a masked avatar in order to remain gender-free. For this exercise, the participants will constantly see the head-gaze ray to guide and ensure the participants will reach the correct head orientation with clean pathway. Finally, we included the important of the sphere (in this case a cube) in the visual biofeedback regarding the references that you suggested.

  1. Only 2 healthy participants that repeated 3 times each angular motion is not enough (6 samples per head orientation). Such a sample requires at least 15 samples (either more participants and/or more repititions) per each of the 12 head orientations. In fact, I did not understand the math because the authors then mention "The volunteers repeated 6 times ...": the sampling count is confusing and needs to be crystal clear!

Authors: Thank you for your remarks. We would like to explain better how we calculated the total sample. First, we followed the previous work of Carnevale et al. (2022) published in Sensors that they validated the controllers of the same HMD device. They analysed only 7 repetitions from the same healthy participant, and we measured 36 repetitions (outcomes per movement, if each axis gives us two movement, for instance, left and right rotation in yaw axis). In order to give a bit of variability in the measurements, we selected two healthy participants instead of one subject. Anyways, we are trying to test the device and the capacity of give good results always (intrarater) regardless of the person who is wearing the device at the time. We would like to know the capacity to detect changes in neck disorders so we will prepare a sample with more people because the variability is more important in this case. Finally, we clarified that sentence to make easier the understanding of the study.

  1. Figure 3: please explain what is the meaning of the horizontal black lines; does the user also see the graphs? please clarify; and again, the color contrast is poor (colored graphs against a bray-blue gradient); This figure needs to be revisited and improved.

Authors: Thank you for your recommendations. The black lines represent 0º of the selected movement. According to the figure, we revisited and improved it according to your suggestions.

  1. Figure 3: Max ROM, Maximum Range of Motion. --> Max ROM - Maximum Range of Motion.

Authors: Thank you for your suggestion. We changed the term according to your suggestion.

  1. 2.12 Data analysis --> sub-section 12 ?!?; please update the section numbers throughout the manuscript.

Authors: Thank you for your remark. We suppose it was a mistake updating the numbers. We have corrected that mistake.

  1. All equations need to be numbered and all equations need to be formated ... they seem that they were writen inline and not with an Equation editor (e.g., the division line should cross the entire right side of the formulas and be represented by a modest '/')

Authors: Thank you for your comment. We checked the Instructions for Authors of the Journal and changed the equations in order to be more appropriate.

  1. limits of agreement (LoA) --> Limits of Agreement (LoA)

Authors: Thank you for your recommendation. We reviewed the text and made all the changes.

Reviewer 3 Report

Patients with neck pathology could have been taken in the study to validate it.

Author Response

Point-by-point response to reviewer 3

Manuscript Number:   sensors-2227091

"Head-Mounted Display for clinical evaluation of neck movement validation with Meta Quest 2"

Dear Editor and reviewer,

Thank you very much for having in mind and review our manuscript. We are grateful for all the advice, comments and corrections suggested, which we believe are right to improve the quality of this paper.

Next, we will try to answer point by point to all your questions and suggestions, as well as specify the changes we have carried out. We highlighted in yellow all the changes in the manuscript.

These variables are assessed with a linear regression analysis.

The manuscript has various important limitations:

  1. Patients with neck pathology could have been taken in the study to validate it.

Authors: Thank you for your recommendation. We agree with your comment. It is necessary future studies in this clinical population as we mentioned in Limitation section. Despite this, the aim of this validation study was analysed the device and the sensor inside to use properly in neck pathologies or other intervention where a move of the head is included. Thus, the use of a healthy volunteers confirm that this technology can be transfered in the rest of clinical populations. We recommend that the next step will be a more specific study in neck pain patients trying to analyse the kinematics as an assessment tool but this work that we performed was the first cornerstone.

Round 2

Reviewer 2 Report

I thank the authors for addressing most of my comments. 

But before recommending the manuscript for publication, I ask them to perform the following tasks:

- in Figure 3, add this sentence at the end of the caption: "The black lines represent 0º of the selected movement."

- all equations need to be formatted with an Equation editor for 'publication quality' sake (e.g., the division line should cross the entire right side of the formulas and not be represented by a modest '/')

- reference 34 should be formatted as "Lopes, D.S.; Faria, A.; Barriga, A.; Caneira, S.; Baptista, F.; Matos, C.; Neves, A.F.; Prates, L.; Pereira, Â .M.; Nicolau, H. Visual Biofeedback for Upper Limb Compensatory Movements: A Preliminary Study Next to Rehabilitation Professionals; The Eurographics Association, 2019; ISBN 978-3-03868-088-8"

Author Response

Point-by-point response to reviewer 2 (Round 2)

Manuscript Number:   sensors-2227091

"Head-Mounted Display for clinical evaluation of neck movement validation with Meta Quest 2"

Dear Editor and reviewer,

Thank you very much for having in mind and review our manuscript. We are grateful for all the advice, comments and corrections suggested, which we believe are right to improve the quality of this paper.

Next, we will try to answer point by point to all your questions and suggestions, as well as specify the changes we have carried out. We highlighted in yellow all the changes in the manuscript.

I thank the authors for addressing most of my comments.

Authors: Thank you for your comment. We appreciate your suggestions from the previous round. The manuscript improved a lot the quality and soundness.

But before recommending the manuscript for publication, I ask them to perform the following tasks:

- in Figure 3, add this sentence at the end of the caption: "The black lines represent 0º of the selected movement."

Authors: Thank you for your suggestion. We added the sentence in Figure 3.

- all equations need to be formatted with an Equation editor for 'publication quality' sake (e.g., the division line should cross the entire right side of the formulas and not be represented by a modest '/')

Authors: Thank you for your remark. We are sorry, we understand totally your suggestion now. We used the Equation editor to improve the presentation of the formulas.

- reference 34 should be formatted as "Lopes, D.S.; Faria, A.; Barriga, A.; Caneira, S.; Baptista, F.; Matos, C.; Neves, A.F.; Prates, L.; Pereira, Â .M.; Nicolau, H. Visual Biofeedback for Upper Limb Compensatory Movements: A Preliminary Study Next to Rehabilitation Professionals; The Eurographics Association, 2019; ISBN 978-3-03868-088-8"

Authors: Thank you for your comment. We corrected the formatting mistake.